# *Lactobacillus plantarum* 17-1 Ameliorates DSS-Induced Colitis by Modulating the Colonic Microbiota Composition and Metabolome in Mice

**DOI:** 10.3390/nu17081348

**Published:** 2025-04-15

**Authors:** Beibei He, Tao Duan, Dandan Hu, Lixian Chen, Lin Qiao, Dan Song, Li Wang, Shijie Fan, Kunru Teng, Weiwei Wang, Aike Li

**Affiliations:** 1Academy of National Food and Strategic Reserves Administration, Beijing 100037, China; hbb@ags.ac.cn (B.H.); dt@ags.ac.cn (T.D.); m18855995768@163.com (D.H.); clx@ags.ac.cn (L.C.); ql@ags.ac.cn (L.Q.); sd@ags.ac.cn (D.S.); wl@ags.ac.cn (L.W.); fsj@ags.ac.cn (S.F.); tkr@ags.ac.cn (K.T.); 2Faculty of Food Science and Engineering, Central South University of Forestry and Technology, Changsha 410004, China

**Keywords:** *Lactobacillus plantarum* 17-1, DSS, inflammatory cytokines, microbiota composition, untargeted metabolome

## Abstract

**Background/Objectives**: *Lactobacillus* strains are widely used as probiotics in the functional food industry and show potential for treating inflammatory bowel disease (IBD). However, the strain specificity and limited stress resistance of *Lactobacillus* restricts its therapeutic effectiveness. The aim of this study was to investigate the effects of dietary supplementation with microencapsulated *Lactobacillus plantarum* 17-1 on the intestinal immune responses, gut microbiota composition, and metabolic characteristics in colitis mice. **Methods**: Mice were pre-fed a diet containing microencapsulated *Lactobacillus plantarum* 17-1 for 3 weeks and then treated with 2.5% dextran sulfate sodium (DSS) in drinking water for 8 days to induce colitis. **Results**: The results showed that microencapsulated *Lactobacillus plantarum* 17-1 effectively alleviated clinical symptoms and histopathological features of colitis mice and suppressed the up-regulation of pro-inflammatory cytokines IL-6 and IL-17 in the colon of colitis mice. Additionally, *Lactobacillus plantarum* 17-1 significantly increased the relative abundance of several beneficial bacterial taxa, including *Ruminococcaceae*_UCG_014, *Bacteroides*, *Prevotellaceae*_UCG_001, *Lactococcus*, *Weissella*, *Pediococcus*, and so on. Moreover, it regulated the levels of multiple inflammation-related metabolites involved in linolenic acid metabolism, arachidonic acid metabolism, primary bile acid biosynthesis, and tyrosine metabolism. **Conclusions**: These results suggest that dietary supplementation with microencapsulated *Lactobacillus plantarum* 17-1 reduced colitis inflammation in mice by modulating the intestinal microbiota composition and metabolic characteristics, which may serve as a potential therapeutic strategy for IBD.

## 1. Introduction

Ulcerative colitis (UC) is a subtype of inflammatory bowel disease (IBD) characterized by chronic ulcerative inflammation of colonic mucosa and submucosa [1]. The pathogenesis of UC is multifactorial, involving genetic predispositions, disruption of the intestinal barrier, and activation of the mucosal immune system [2,3]. Specifically, the disruption of homeostasis between intestinal microbiota and the mucosal immune system is emerging as a major contributing factor to the development and progression of UC. For example, intestinal microbiota dysregulation in UC patients is often manifested by a decrease in beneficial bacteria (such as *bifidobacterium* and *Lactobacillus*) and an increase in harmful bacteria (such as *Enterobacteriaceae*), resulting in decreased levels of intestinal microbial metabolites that regulate host immune responses (such as short-chain fatty acids (SCFAs), bile acids, amino acids, and lipid metabolites) [4,5,6]. And the impaired mucosal barrier in UC patients may result in a breakdown of the physical isolation between microbiota and the intestinal immune system, facilitating bacteria access to intestinal epithelial and thereby eliciting an abnormal immune response [7]. Traditionally used drugs to treat IBD include 5-aminosalicylic acid, steroids, and immunosuppressants, which have serious side effects (such as leucopenia, pancreatitis, and increased risk of malignancy) and are not suitable for long-term treatment [8,9]. In recent years, some probiotics, when administered at effective doses, have been proved to protect against UC in both animal model and human studies [10,11].

*Lactobacillus plantarum* (*L. plantarum*) is a lactic acid bacterium widely used in the food and dairy industries. It is known to have effective capacity in UC therapy through suppressing pro-inflammatory cytokine expressions, enhancing the intestinal epithelial barrier, as well as modulating gut microbiota [12]. However, the beneficial effects of *L. plantarum* on UC patients rely on the heterogeneity among different strains [13]. Additionally, due to the influence of gastrointestinal environmental factors such as gastric acid and bile salts, the limited stress resistance of *L. plantarum* makes it difficult to achieve effective concentrations at the disease site, thus affecting its therapeutic effect [14]. Moreover, the specific effects of *L. plantarum* on the gut microbiota and their metabolites, as well as the mechanisms underlying its ability to alleviate UC, are still not fully understood.

Dextran sulfate sodium (DSS)-induced colitis in mice can simulate the key pathological features of human UC and is a common model for studying the anti-inflammatory effects of probiotics. The aim of this study was to investigate: (1) the effect of *L. plantarum* 17-1 on clinical symptoms and histopathological features of colitis mice; (2) the effect of *L. plantarum* 17-1 on the levels of inflammation-related cytokines in colon tissue; and (3) the regulatory effect of *L. plantarum* 17-1 on intestinal microbial composition and metabolic characteristics. We hypothesized that microencapsulated *L. plantarum* 17-1 could significantly improve the symptoms of DSS-induced colitis in mice and inhibit the inflammatory response by modulating the gut microbiome and metabolome.

## 2. Materials and Methods

### 2.1. Materials

*Lactobacillus plantarum* 17-1 was isolated from naturally fermented dairy products in China and identified through 16S rDNA similarity analysis. The strain was preserved and cultured anaerobically in MRS broth (02-293X; AOBOXING Bio-tech Co., Ltd., Beijing, China) at 37 °C for 24 h. Microencapsulation of *L. plantarum* 17-1 was conducted using a modified version of the emulsion method described by Qi et al. [15]. Briefly, *L. plantarum* 17-1 was first activated twice in MRS broth at 37 °C under anaerobic conditions. The bacterial suspension (700 mL, 1 × 10⁹ CFU/mL) was mixed with 5 L of sterile 2.0% sodium alginate (S11053; Yuanye Bio-Technology Co., Ltd., Shanghai, China) solution and 75 g of calcium carbonate (10005717; Sinopharm Chemical Reagent Co., Ltd., Shanghai, China) and emulsified into paraffin oil (30139828; Sinopharm Chemical Reagent Co., Ltd., Shanghai, China) containing 0.2% Span 80 (30189828; Sinopharm Chemical Reagent Co., Ltd., Shanghai, China) using mechanical stirring (400 rpm, 5 min). 90 mL Glacial acetic acid (10000208; Sinopharm Chemical Reagent Co., Ltd., Shanghai, China) was then added to release calcium ions from calcium carbonate, inducing alginate gelation. After 10 min, 200 mL of saturated calcium chloride (53219961; Sinopharm Chemical Reagent Co., Ltd., Shanghai, China) solution was added to harden the microbeads. Subsequently, the oil phase was removed, and the solidified microbeads were washed with cold saline. Encapsulated cells were cultured in MRS medium (70 L, 37 °C, 20 h) to enhance viability. Finally, microcapsules were harvested by centrifugation, blended with corn starch (S11149; Yuanye Bio-Technology Co., Ltd., Shanghai, China), and dried in a fluidized bed dryer.

### 2.2. Animal Treatments

The experimental protocol was approved by the Animal Ethics Committee of the Academy of National Food and Strategic Reserves Administration (Approval No. 2019M06; Date: 12 May 2024).

A total of 48 specific pathogen-free (SPF) male BALB/c mice (5-week-old, 18–22 g) were obtained from HFK Bioscience Co., Ltd. (Beijing, China). Animals were acclimatized for 7 days in individually ventilated cages (IVC, 4 mice/cage) under controlled conditions (23 ± 1 °C, 50 ± 5% humidity, 12 h light/dark cycle), with ad libitum access to feed (AIN-93G diet) and water. Based on the feasibility of the experimental design and the experience of the preliminary study, mice were allocated to 4 weight-matched groups (*n* = 12): CON (basal diet + distilled water), DSS (basal diet + 2.5% (*w*/*v*) DSS (MW 36–50 kDa, 02160110-CF; MP Biomedicals, Beijing, China) in drinking water), LP (basal diet + 10⁷ CFU/g *L. plantarum* 17-1), and LP + DSS (LP treatment + DSS induction). DSS solution was prepared daily and provided from day 22 to day 29, until DSS-treated mice exhibited visible or occult fecal blood. On day 29, mice were euthanized by cervical dislocation following 12 h fasting, and colon tissue and content were collected for subsequent analysis (Figure 1A). Experimental operators were unblinded to administer treatments, while outcome assessors and statisticians remained blinded through pseudonymized sample coding.

### 2.3. Evaluation of the Disease Activity Index and Histopathology

The disease activity index (DAI) scores of mice from day 22 to day 29 were calculated based on body weight (BW) loss, stool consistency, and the presence of blood in stool [16]. Colon length was measured from the ileum–cecal junction to the proximal rectum. Histopathological analysis was performed according to the method by Xia et al. [17]. Briefly, 1 mm sections of colon tissue were fixed in 4% paraformaldehyde (BL539A; Labgic Technology Co., Ltd., Beijing, China) for 24 h, followed by progressive ethanol dehydration, xylene (10023418; Sinopharm Chemical Reagent Co., Ltd., Shanghai, China) clearing, paraffin embedding, sectioning, and staining with hematoxylin and eosin (HE) (R20570; Yuanye Bio-Technology Co., Ltd., Shanghai, China). The stained sections were qualitatively analyzed by two blinded observers via light microscopy (Nikon Eclipse E400, 200×; Tokyo, Japan) to identify characteristic injury patterns. Representative images were captured with a Canon EOS 6D camera (Tokyo, Japan) under standardized settings.

### 2.4. Measurement of Colon Cytokines

Colon tissues were weighed and homogenized in ice-cold phosphate-buffered saline (PBS) at a ratio of 1:9 (*w*/*v*). The homogenates were then centrifuged at 5000× *g* for 10 min at 4 °C. The resulting supernatants were analyzed to quantify the concentrations of tumor necrosis factor alpha (TNF; E-EL-M3063), lymphotoxin-α (E-EL-M1210), interleukin-1β (IL-1β; E-EL-M0037), interleukin-6 (IL-6; E-EL-R0015), interleukin-10 (IL-10; E-EL-M0046), and interleukin-17 (IL-17; E-EL-M0047) using commercial enzyme-linked immunosorbent assay (ELISA) kits, following the manufacturer’s instructions (Elabscience Biotechnology Co., Ltd., Wuhan, China).

### 2.5. 16S rRNA Gene Sequencing and Microbial Analysis

#### 2.5.1. DNA Extraction and PCR Amplification

Genomic DNA from the microbial community present in the colonic content was isolated utilizing the E.Z.N.A.^®^ Stool DNA Kit (D4015; Omega Bio-tek, Norcross, GA, USA) following the manufacturer’s protocol. The integrity of the extracted genomic DNA was assessed via agarose gel electrophoresis, while its concentration and purity were measured using a NanoDrop 2000 UV-Vis spectrophotometer (Thermo Scientific, Wilmington, NC, USA). The V3-V4 region of the bacterial 16S rRNA genes was amplified using primers 338F (5′-ACTCCTACGGGAGGCAGCAG-3′) and 806R (5′-GGACTACHVGGGTWTCTAAT-3′) in a PCR thermocycler (GeneAmp 9700; ABI, Foster City, CA, USA). The PCR conditions were as follows: 95 °C for 3 min; 27 cycles of 95 °C denaturation for 30 s, 55 °C annealing for 30 s, and 72 °C extension for 45 s; and a final extension at 72 °C for 10 min [18].

#### 2.5.2. Illumina Sequencing and Data Processing

The resulting PCR products were purified with the AxyPrep DNA Gel Extraction Kit (AP-GX; Axygen Biosciences, Union City, CA, USA) and quantified using a Quantus™ E6150 Fluorometer (Promega, Madison, WI, USA). Subsequently, the purified amplicons were pooled in equimolar concentrations and subjected to paired-end sequencing on an Illumina MiSeq platform (Illumina, San Diego, CA, USA) in accordance with the standard protocols at Majorbio Bio-Pharm Technology Co. Ltd. (Shanghai, China).

A total of 6,987,098 raw reads were acquired and subsequently merged using FLASH software version 1.2.7 (SourceForge, San Francisco, CA, USA) [19]. Quality filtering was performed with fastp version 0.20.0 (OpenGene, Shenzhen, China) [20]. Only sequences with an overlap longer than 10 bp and without any mismatch were assembled according to their overlap sequence. Following the removal of chimeric sequences, operational taxonomic units (OTUs) were clustered with 97% similarity using UPARSE software version 7.1 (Drive5, Austin, TX, USA) [21]. The Ribosomal Database Project (RDP) Classifier (version 2.14; Michigan State University, East Lansing, MI, USA) was used to analyze the phylogenetic affiliation of each 16S rRNA gene sequence with confidence greater than 70%. The Principal Coordinate Analysis (PCoA) based on unweighted UniFrac distance metrics was conducted according to the matrix of distance using the JMP software of SAS (version 8.0.2; SAS Institute, Cary, NC, USA). QIIME software version 1.7.0 (University of Colorado at Boulder, Boulder, CO, USA) was employed to assess α- and β-diversity. Venn diagrams were generated to visualize shared and unique OTUs among experimental groups using the VennDiagram package in R software (version 3.3.1; R Foundation for Statistical Computing, Vienna, Austria). Taxonomic compositions and their relative abundances at phylum and genus were created and visualized using R software. One-way ANOVA with Tukey’s post-hoc tests was implemented in R software to identify differentially abundant genera (*p* < 0.05) across experimental groups. The linear discriminant analysis effect size (LEfSe) method was utilized to identify biomarkers that characterize differences in abundant bacterial taxa between groups. The associations between bacterial taxa and colitis-related indices were evaluated using Spearman’s rank correlation coefficients, with significance determined at *p* < 0.05, and were calculated with R software.

### 2.6. Untargeted Metabolome Profiling Analysis

#### 2.6.1. Metabolite Extraction

Metabolic profiling of colonic content was performed utilizing an ultra-high-performance liquid chromatography–mass spectrometry (UHPLC-MS) methodology, as described by Yao et al. [22], with subsequent analysis conducted via the Majorbio Cloud Platform (www.majorbio.com). Specifically, 50 mg of colonic content (50 mg) was combined with 400 μL of a methanol/water solution (4:1, *v*/*v*) and 20 μL of an internal standard solution (2-chloro-l-phenylalanine (C9294; Sigma-Aldrich, St. Louis, MO, USA) in acetonitrile (A998-4; Fisher Scientific, Pittsburgh, PA, USA), 0.3 mg/mL). The mixture was then flash-frozen in liquid nitrogen, ground using a high-throughput tissue crusher (Scientz-48; NingBo Scientz Biotechnology Co., Ltd., Ningbo, China), and ultrasonically homogenized in ice-cold PBS buffer. Following centrifugation at 13,000× *g* at 4 C for 15 min, the supernatant was collected and transferred into sample vials for UHPLC-MS analysis. A quality control (QC) sample was generated by pooling aliquots from each individual sample.

#### 2.6.2. UHPLC-MS/MS Analysis

The LC-MS/MS analysis of samples was conducted on a Thermo UHPLC-Q Exactive system equipped with an ACQUITY HSS T3 column (100 mm × 2.1 mm i.d., 1.8 μm; Waters, Milford, MA, USA) at Majorbio Bio-Pharm Technology Co., Ltd. (Shanghai, China). The mobile phases were composed of solvent A (0.1% formic acid (F0507; Sigma-Aldrich, St. Louis, MO, USA) in water: acetonitrile, 95: 5, *v*/*v*) and solvent B (0.1% formic acid in acetonitrile: isopropanol (34863; Sigma-Aldrich, St. Louis, MO, USA): water, 47.5: 47.5: 5, *v*/*v*). The flow rate was maintained at 0.40 mL/min, and the column temperature was set at 40 °C.

Mass spectrometric data were acquired using a Thermo UHPLC-Q Exactive HF-X Mass Spectrometer (Thermo Fisher Scientific, Waltham, MA, USA) equipped with an electrospray ionization (ESI) source operating in both positive and negative modes. The optimized instrumental conditions were set as follows: source temperature, 425 °C; sheath gas flow rate, 50 arb; auxiliary gas flow rate, 13 arb; and ion-spray voltage floating, −3500 V in negative mode and 3500 V in positive mode. The normalized collision energy was set to a rolling schedule of 20, 40, and 60 V for MS/MS. The resolution was 60,000 for full MS scans and 7500 for MS/MS scans. Data acquisition was performed in Data Dependent Acquisition (DDA) mode, with a mass range of 70–1050 m/z for detection.

#### 2.6.3. Data Analysis

The initial data were pre-processed using Progenesis QI version 2.3 (Waters, Milford, MA, USA) and uploaded into the MetaboLights database (accession number: MTBLS12244). Metabolite annotation was performed utilizing publicly available biochemical databases, including the Human Metabolome Database (HMDB, version 5.0, http://www.hmdb.ca/, accessed on 20 October 2024; The Metabolomics Innovation Centre, Edmonton, Canada), METabolite LIdentity (METLIN, https://metlin.scripps.edu/, accessed on 21 October 2024; Scripps Research, La Jolla, CA, USA), and a proprietary untargeted database from Majorbio (Shanghai, China). To assess the variations in metabolic compounds across different groups, partial least-squares discrimination analysis (PLS-DA) was conducted using the ropls R package (version 1.6.2; Bioconductor, Cambridge, MA, USA). The variable importance in the projection (VIP) scores was computed within the PLS-DA model. *p*-values were calculated using paired Student’s *t*-tests to compare metabolite concentrations across the groups. Metabolites with a VIP score greater than 1 and a *p*-value less than 0.05 were identified as significantly differential metabolites within each group. These differential metabolites were verified through the Kyoto Encyclopedia of Genes and Genomes (KEGG) database (https://www.kegg.org/; Kanehisa Laboratories, Kyoto, Japan) and cross-referenced with KEGG pathways. Significantly enriched pathways were validated using Fisher’s exact test, implemented via the scipy.stats package in Python (version 3.12.0; Python Software Foundation, New York, NY, USA). Pearson’s correlations between the differentially abundant cecal microbiota metabolites across groups were analyzed using R software.

### 2.7. Statistical Analysis

Values are expressed as the mean ± standard deviation (SD). The statistically significant differences among group means were determined using one-way ANOVA, followed by the Student–Newman–Keuls comparison test of SAS (version 8.0.2, SAS Institute, Cary, NC, USA). Values of *p* < 0.05 were considered statistically significant.

## 3. Results

### 3.1. L. plantarum 17-1 Ameliorated Clinical Features and Intestinal Injury in DSS-Induced-Colitis Mice

During the 3-week pre-treatment phase, no significant difference in BW was observed between the control and the *L. plantarum* treatment group (*p* > 0.05), except for higher feed intake in the *L. plantarum* treatment group compared to the control group (*p* < 0.05, Figure 1B). From the fourth day of DSS exposure, the mice in the DSS group showed significant reduction in BW compared to those in the CON and LP groups (*p* < 0.05, Figure 1C). Additionally, a significant increase in DAI score was observed on the sixth day of DSS exposure in the DSS and DSS + LP groups compared to the CON and LP groups (*p* < 0.05), with a subsequent decrease noted on the eighth day of DSS exposure in the LP + DSS group relative to the DSS group (*p* < 0.05, Figure 1D). Furthermore, colon length was significantly reduced in the DSS group compared to the CON and LP groups (*p* < 0.05), with *L. plantarum* 17-1 supplementation having restored the colon length of the DSS + LP group to a level comparable to that of the CON and LP groups (*p* > 0.05, Figure 1F).

To observe the effects of *L. plantarum* 17-1 on pathological changes in the colonic tissue of mice, HE staining was performed. As shown in Figure 1E, DSS-induced colon injury was characterized by partial destruction of the epithelial structure, damage to the crypt structure, fibrous hyperplasia, and submucosal edema, accompanied by inflammatory cell infiltration. In contrast, *L. plantarum* 17-1 supplementation significantly alleviated these pathological changes caused by DSS treatment, as it repaired the epithelial architecture and reduced inflammatory cell infiltration in the colon tissue.

### 3.2. L. plantarum 17-1 Regulated the Production of Cytokines in the Colon of DSS-Induced-Colitis Mice

The effect of DSS treatment and *L. plantarum* 17-1 supplementation on cytokine levels in colon tissue is shown in Figure 2. Compared with the control group, the colonic levels of pro-inflammatory cytokines TNF, lymphosin-α, IL-1β, IL-6, and IL-17 were significantly increased, while the levels of anti-inflammatory cytokine IL-10 were decreased in the mice of the DSS group (*p* < 0.05). It is worth mentioning that the colonic levels of TNF and IL-6 were decreased, and the level of IL-10 was increased in the mice of the LP group compared to those observed in the control (*p* < 0.05). Notably, administration of *L. plantarum* 17-1 to DSS-exposed mice resulted in a significant reduction in IL-6 and IL-17 levels (*p* < 0.05) compared to the DSS group.

### 3.3. L. plantarum 17-1 Changed Microbiota Diversity in the Colon Digesta of DSS-Induced-Colitis Mice

To elucidate the response of gut microbiota in DSS-induced-colitis mice to a diet containing *L. plantarum* 17-1, microbiota composition was assessed using high-throughput 16S rRNA gene sequencing. As shown in Figure 2A, the Sobs index of bacterial communities was significantly reduced within the colonic digesta of mice in the DSS group, compared to that of the CON group (*p* < 0.05). However, administration of *L. plantarum* 17-1 mitigated this reduction, resulting in a significant increase (*p* < 0.05) in the Sobs index of bacterial communities in the DSS + LP group, compared to the DSS group (*p* < 0.05). The Shannon index of bacterial communities within the colonic digesta of mice in the LP group was higher compared to the DSS group (*p* < 0.05), while there were no significant differences among the CON, DSS, and DSS + LP groups (*p* > 0.05). Based on unweighted UniFrac distances, PCoA demonstrated a clear separation of microbiota profiles in the colonic digesta between the four treatment groups (R = 0.2405, *p* = 0.001, ANOSIM, Figure 3B).

### 3.4. L. plantarum 17-1 Induced Shift in Gut Microbiota Composition in DSS-Induced-Colitis Mice

The microbiota composition was analyzed at both the phylum and genus levels (Figure 3C). At the phylum level, the colonic microbiota of mice in all the groups was predominantly composed of Firmicutes, Bacteroidetes, and Actinobacteria. At the genus level, OTUs associated with *Lactobacillus*, *Ruminococcaceae*_UCG_014, norank_f_Muribaculaceae, *Lachnoclostridium*, *Lachnospiraceae*_NK4A136_group, norank_f_Lachnospiraceae, unclassified_f_Lachnospiraceae, *Enterorhabdus*, Bacteroides, and *Prevotellaceae*_UCG_001 were predominant in the colonic digesta of mice in all the groups. The Venn diagram showed that there were 384 common OTUs in all of the four treatment groups, and the numbers of unique OTUs in the CON, DSS, LP, and LP + DSS groups were 19, 11, 39, and 25, respectively. The number of common OTUs in both the CON and the LP group was 19, the number of common OTUs between the CON and the LP + DSS group was 7, while there was only 1 common OTU in the CON and DSS groups (Figure 3D). A one-way ANOVA analysis was used to test the differentially abundant genera with relative abundance greater than 1% among the four treatment groups (Figure 3E). The results showed that the relative abundance of norank_f_Muribaculaceae was higher in the CON group compared to the DSS and LP + DSS groups (*p* < 0.05), and the relative abundance of norank_f_Lachnospiraceae and *Roseburia* was higher in the LP group compared to the DSS and LP + DSS groups (*p* < 0.05), while the relative abundance of *Ruminiclostridium*_6 was higher in the DSS group compared to the LP group (*p* < 0.05).

LEfSe analysis was conducted to identify biomarkers that characterize the differences in the abundances of bacterial taxa between groups, as depicted in Figure 4. The relative abundances of six genera, including *Lachnospiraceae*_UCG_006, *Ruminiclostridium*_6, *Escherichia*_Shigella, *Papillibacter*, *Staphylococcus*, and *Catabacter*, were significantly higher (*p* < 0.05) in the colonic digesta of mice in the DSS group. Conversely, the relative abundances of eight genera, including *Ruminococcaceae*_UCG_014, *Bacteroides*, *Prevotellaceae*_UCG_001, *Lactococcus*, *Lachnospiraceae*_FCS020_group, *Weissella*, *Eubacterium*_fissicatena_group, and *Pediococcus*, were significantly higher (*p* < 0.05) in mice from the LP + DSS group. Additionally, 21 genera, such as *Candidatus*_Saccharimonas, *Eubacterium*_xylanophilum_group, and *Roseburia*, exhibited a significantly higher abundance (*p* < 0.05) in mice from the LP group.

### 3.5. Spearman Correlation Analysis of Differentially Abundant Genera and Colitis Injury Indices

To further clarify the relationship between colitis injury indices and shifts in gut microbiota composition, correlation analysis between injury indices and the top 20 genera with significant changes in relative abundance was conducted across all of the four treatment groups (Figure 5). The genera norank_f_Lachnospiraceae, *Eubacterium*_xylanophilum_group, *Roseburia*, *Enterorhabdus*, *Candidatus*_Saccharimonas, and *Alistipes* exhibited significant negative correlations with inflammatory cytokines TNF, lymphotoxin-α, IL-6, and IL-17, while they showed positive correlations with anti-inflammatory cytokine IL-10, BW change, and colon length in mice. Conversely, the genera *Ruminiclostridium*_6, *Lactobacillus*, and *Akkermansia* demonstrated positive correlations with lymphotoxin-α, IL-6, and IL-17, while showing negative correlations with cytokine IL-10, BW change, and colon length in mice.

### 3.6. L. plantarum 17-1 Altered Gut Metabolic Profiles in DSS-Induced-Colitis Mice

To assess the impact of *L. plantarum* 17-1 on gut metabolic profiles in DSS-induced-colitis mice, colonic digesta were analyzed using UHPLC-MS across the CON, DSS, and LP + DSS groups. PLS-DA demonstrated significant differentiation in metabolic profiles between the CON and DSS groups (R^2^ = 0.9654, Q^2^ = 0.273; Figure 6A), as well as the DSS and LP + DSS groups (R^2^ = 0.9699, Q^2^ = 0.2168; Figure 6B).

The selection of altered metabolites between groups was based on a threshold of VIP values greater than 1 from the PLS-DA model and a *p*-value < 0.05 from Student’s *t*-test. A total of 20 differential metabolites between the DSS and CON groups were annotated in the KEGG database, of which 14 metabolites (5-O-methylembelin, L-homocystine, L-homoserine, glycocholic acid, bile acid, 7-oxodeoxycholate, 10-oxodecanoate, D-aspartic acid, 1-palmitoylglycerol 3-phosphate, 3β, 7α-Dihydroxy-5-cholestenoate, murideoxycholic acid, (S)-10,16-dihydroxyhexadecanoic acid, 9,10,18-trihydroxystearate, α-linolenic acid) were decreased, and 6 metabolites (4-maleylacetoacetic acid, 3-ketosphingosine, 6-hydroxynicotinate, cholestenone, cellobiose, 5-acetamidopentanoate) were increased in the DSS group compared to the CON group (Appendix A). Among these differential metabolites, eight metabolites with VIP values above 1.5 are shown in Figure 7A. Significantly enriched pathways analysis showed that α-linolenic acid was involved in the α-linolenic acid metabolism pathway; glycocholic acid, cholic acid, and 3β,7α-dihydroxy-5-cholestenoate were involved in the primary bile acid biosynthesis pathway; 4-maleylacetoacetate was involved in the tyrosine metabolism pathway; and D-aspartic acid was involved in the D-amino acid metabolism pathway (Figure 7B). In the comparison between the DSS and LP + DSS groups, 24 differential metabolites were annotated in the KEGG database, of which 7 metabolites ((9Z)-(7S,8S)-dihydroxyoctadecenoic acid, 4-maleylacetoacetic acid, D-(+)-malic acid, (9Z,11E)-octadecadienoic acid, arachidonic acid, cellobiose, anandamide) were decreased and 17 metabolites (N-acetyl-L-phenylalanine, D-urobilinogen, suberic acid, murideoxycholic acid, ethyl (E,Z)-decadienoate, 9,10,13-triHOME, ophiobolin A, (9R,10R)-dihydroxyoctadecanoic acid, (9Z)-(13S)-12,13-epoxyoctadeca-9,11-dienoic acid, 9,10-DHOME, (9Z,12Z)-(8R)-hydroxyoctadeca-9,12-dienoic acid, tetradecanedioic acid, (4Z,7Z,10Z,13Z,16Z,19Z)-docosahexaenoic acid, acetyl-DL-leucine, ecklonialactone A, 5-O-methylembelin, 3β-Hydroxy-4β-methyl-5α-cholest-7-ene-4α-carboxylate) were increased in the LP + DSS group compared to the DSS group (Appendix A), and the 9 metabolites with VIP values above 1.5 are shown in Figure 7C. Significantly enriched pathways analysis showed that arachidonic acid is involved in the arachidonic acid metabolism pathway, 4-maleylacetoacetate is involved in the tyrosine metabolism pathway, and ecklonialactone A is involved in the primary bile acid biosynthesis pathway (Figure 7D).

### 3.7. Spearman Correlation Analysis of Differentially Abundant Genera and Metabolites

Furthermore, we conducted a correlation analysis between significantly changed genera and significantly changed metabolites with VIP values above 1.5 across the CON and DSS groups, as well as the LP + DSS and DSS groups. In the comparison between the CON and DSS groups, 3-Ketosphingosine exhibited positive correlations with *Ruminiclostridium*_6, *Escherichia*_Shigella, and *Papillibacter*. Conversely, L-homoserine, bile acid, and 7-oxodeoxycholate demonstrated negative correlations with these same genera (refer to Figure 8A). In the analysis between the DSS and LP + DSS groups, N-acetyl-L-phenylalanine, suberic acid, and ethyl(E,Z)-decadienoate were positively correlated with *Bacteroides* and *Pediococcus*, whereas (Z)-(7S,8S)-dihydroxyoctadecenoic acid and D-(+)-malic acid were negatively correlated with these genera (refer to Figure 8B).

## 4. Discussion

An increasing amount of research has indicated that *L. plantarum* holds significant potential to restore disrupted intestinal flora and serve as an effective probiotic to ameliorate colitis [23,24]. However, due to the limited stress resistance of *L. plantarum* to gastrointestinal environmental factors such as gastric acid and bile salts, it is difficult to achieve an effective concentration in the colon, thus affecting its efficacy in the treatment of colitis. In the present study, we employed a microencapsulation technique for *L. plantarum* 17-1 and incorporated it into the diet of mice, which not only enhanced the survivability of *L. plantarum* 17-1 through the harsh conditions of the gastrointestinal tract, but also ensured sustained release within the colon, where its therapeutic effects are most needed. The results demonstrated that dietary supplementation of microencapsulated *L. plantarum* 17-1 effectively alleviated disease symptoms and pathological damage, modulated inflammatory cytokines, and maintained the dynamic equilibrium of intestinal flora in mice with DSS-induced colitis.

Prior to DSS treatment, mice were pre-fed with a probiotic-supplemented diet for 3 weeks, which resulted in increased food intake. Although no changes in body weight were observed, this early intervention phase underscores the importance of establishing a favorable gut environment before the onset of disease [25,26]. DSS-induced colitis is usually characterized by weight loss, shortened colon, and fecal bleeding, which was also observed in the present study. However, administration of *L. plantarum* 17-1 demonstrated a mitigating effect on these episodic symptoms and exhibited synergistic benefits (Figure 1).

Clinical studies consistently show that the severity of IBD is strongly correlated with the balance between pro-inflammatory and anti-inflammatory cytokines [27]. In the present study, the colonic levels of pro-inflammatory cytokines TNF, lymphosin-α, IL-1β, IL-6, and IL-17 were significantly increased, while the levels of anti-inflammatory cytokines IL-10 were decreased in the mice of the DSS group (Figure 2), which further validated the critical role of inflammatory cytokines in maintaining intestinal homeostasis [28,29,30]. It has been proved that *L. plantarum* can interact with gut-associated lymphocytes and epithelial cells to reduce pro-inflammatory cytokine levels while increasing anti-inflammatory cytokine secretion [31,32]. Furthermore, the ability of *L. plantarum* to interact with gut epithelial cells reinforces its role as a key player in gut barrier function and immune modulation [33,34]. When treated with a diet supplemented with *L. plantarum* 17-1, the levels of IL-6 and IL-17 were reduced in DSS-induced-colitis mice (Figure 2D,F), which may help restore the disrupted balance in IBD patients, potentially alleviating symptoms and promoting recovery.

Although the exact mechanism is still unclear, it is widely believed that the balance between the gut microbiota and the mucosal immune system plays a key regulatory role in IBD development [35]. In our study, we observed elevated levels of specific bacterial taxa in colitis mice, including *Lachnospiraceae*_UCG_006, *Ruminiclostridium*_6, *Escherichia*_Shigella, *Papillibacter*, *Staphylococcus*, and *Catabacter* (Figure 5). Notably, the relative abundance of *Ruminiclostridium*_6 demonstrated a positive correlation with lymphosin-α and IL-17, and a negative correlation with BW change, suggesting its potential as an indicator of colitis progression [36]. Our results also identified a strong positive correlation between *Escherichia*_Shigella and pro-inflammatory cytokines IL-17 and IFN-γ, which aligns with previous studies highlighting the pathogenic nature of *Escherichia*_Shigella in colitic mice [37]. The reduction in the relative abundance of *Escherichia*_Shigella and down-regulation of pro-inflammatory cytokines following *L. plantarum* treatment underscores its therapeutic potential, which is consistent with other studies demonstrating the beneficial effects of *L. plantarum* in decreasing gut pathogenic bacteria and mitigating inflammatory responses [38,39]. Additionally, our data revealed a significant increase in the relative abundance of *Staphylococcus* following DSS treatment, which is consistent with other research showing that certain *Staphylococcus* species can promote intestinal inflammation through toxin production or immune system modulation [40]. Additionally, we noted a significant increase in the phylum Verrucomicrobia in colitic mice (Figure 3C), primarily due to the proliferation of *Akkermansia*, which was positively correlated with inflammatory cytokines and negatively correlated with anti-inflammatory cytokines as well as colon length (Figure 5). *Akkermansia* expresses a wide range of enzymes, including glycosyl hydrolases, proteases, and sialidases, which enable it to break down mucin glycoproteins [41,42].

Probiotics, as live microorganisms, play a pivotal role in modulating gut microbiota composition and restoring the mucosal immune responses disrupted by chronic intestinal inflammation [43]. In this study, we demonstrated that *L. plantarum* 17-1 enhanced the relative abundance of several beneficial genera, including *Ruminococcaceae*_UCG_014, *Bacteroides*, *Prevotellaceae*_UCG_001, *Lactococcus*, *Lachnospiraceae*_FCS020_group, *Weissella*, *Eubacterium*_fissicatena_group, and *Pediococcus* (Figure 4). The effect of *Ruminococcaceae*_UCG_014 in producing SCFA and mitigating colonic barrier dysfunction and inflammation were observed in several disease models [44,45,46]. Notably, *Bacteroides* are known to enhance intestinal barrier function by promoting the expression of intestinal tight junction proteins and reducing apoptosis of epithelial cells [47,48], and the increase in *Bacteroides* in the LP + DSS group suggests that *L. plantarum* 17-1 may restore the intestinal microbiota to maintain the stability and function of the internal environment. *Bacteroidetes* and *Prevotellaceae*_UCG-001, both belonging to the phylum Bacteroidetes, may degrade polysaccharides and supply nutrients to other probiotics for producing anti-inflammatory metabolites [49]. The immunomodulatory and mucosal repair properties of *Lactococcus*, *Weissella*, and *Pediococcus* were also widely concluded in several colitis models [50,51,52,53].

Metabolites are the main substances that microorganisms use to regulate the local microenvironment of the intestine and thus affect the function of the organisms. In the present study, the level of α-linolenic acid was decreased in the DSS group compared to the CON group, which was involved in the α-linolenic acid metabolism pathway. Linolenic acid is an important polyunsaturated fatty acid, and its metabolites may be involved in regulating the production of inflammatory mediators and cell signaling [54]. Studies have shown that changes in metabolites in the linolenic acid metabolic pathway are closely related to the severity of inflammation in mouse models of DSS-induced colitis. By regulating the metabolism of linolenic acid, the symptoms of DSS-induced colitis can be significantly reduced [55]. The level of 4-maleylacetoacetate was increased in the DSS group compared to the CON group, while decreased in the LP + DSS group compared to the DSS group (Figure 7). 4-maleylacetoacetate is involved in the tyrosine metabolism pathway, changes in which may influence the development of DSS by modulating intestinal barrier function, immune cell signaling, and inflammatory responses [56,57]. Probiotics may indirectly affect tyrosine metabolism in DSS-induced colitis by regulating the composition and function of the gut microbiota. Arachidonic acid was decreased in the LP + DSS group compared to the DSS group, which is involved in the arachidonic acid metabolism pathway. In a DSS-induced UC model, arachidonic acid metabolism is one of the main metabolic pathways affected, and it is closely related to the regulation of inflammatory response [58].

The level of 3-ketosphingosine was increased in the DSS group compared to the CON group, and exhibited positive correlations with *Ruminiclostridium*_6, *Escherichia*_Shigella, and *Papillibacter* (Figure 8A). It acts as an intermediate of sphingolipid metabolism and may affect intestinal health by regulating cell signaling pathways and inflammatory response [59]. Studies have shown that gut microbiota are able to convert primary bile acids into secondary bile acids, which are strongly associated with colon inflammation [60]. In addition, some microorganisms such as *Escherichia*_Shigella may participate in the metabolism of bile acids, thus affecting the inflammatory response, which is consistent with our results that bile acid was down-regulated in the DSS group compared to the CON group [61]. L-homoserine and 7-oxodeoxycholate are metabolites that are involved in amino acid and bile acid metabolism. While there is currently a lack of direct research on the role of these metabolites in DSS models, their potential role in gut health has been widely demonstrated [59,62,63,64].

## 5. Conclusions

In summary, microencapsulated *Lactobacillus plantarum* 17-1 demonstrated significant therapeutic efficacy in ameliorating clinical manifestations and inflammatory responses in DSS-induced-colitis mice. This protective effect was mediated through restoration of gut microbial homeostasis and modulation of colitis-associated metabolic profiles.

## Figures and Tables

**Figure 1 nutrients-17-01348-f001:**
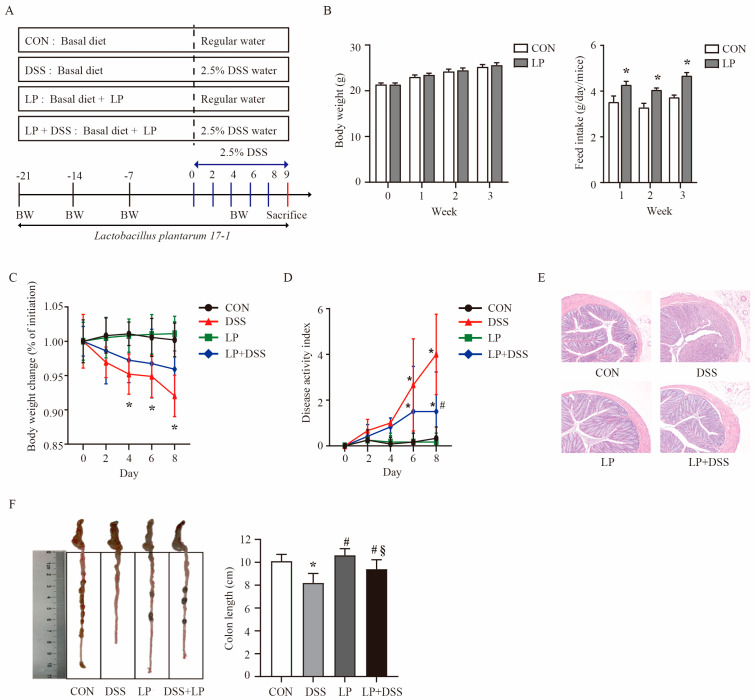
Improvement of the apparent characteristics of DSS-induced colitis by *Lactobacillus plantarum* 17-1. Data are expressed as the mean ± SD (*n* = 12). Values with different superscripts are significantly different within the same level. * *p* < 0.05, compared with the control group; # *p* < 0.05, compared with the DSS group; § *p* < 0.05, compared with the LP group. (**A**) Experimental design; (**B**) body weight and food intake before DSS treatment; (**C**) body weight change after DSS treatment; (**D**) DAI after DSS treatment; (**E**) histopathological analysis; (**F**) colon length.

**Figure 2 nutrients-17-01348-f002:**
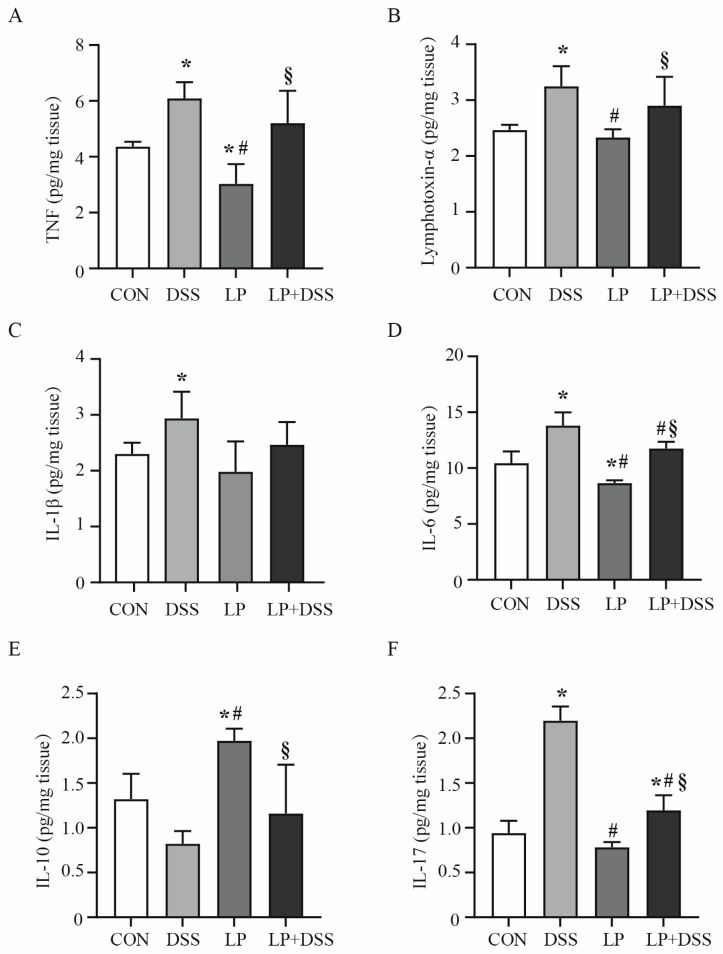
Cytokine levels in colon tissue of mice. Data are expressed as the mean ± SD (*n* = 8). Values with different superscripts are significantly different within the same level. * *p* < 0.05, compared with the control group; # *p* < 0.05, compared with the LP group; *§ p* < 0.05, compared with the DSS group. (**A**) The levels of TNF; (**B**) the levels of lymphosin-α; (**C**) the levels of IL-1β; (**D**) the levels of IL-6; (**E**) the levels of IL-10; (**F**) the levels of IL-17.

**Figure 3 nutrients-17-01348-f003:**
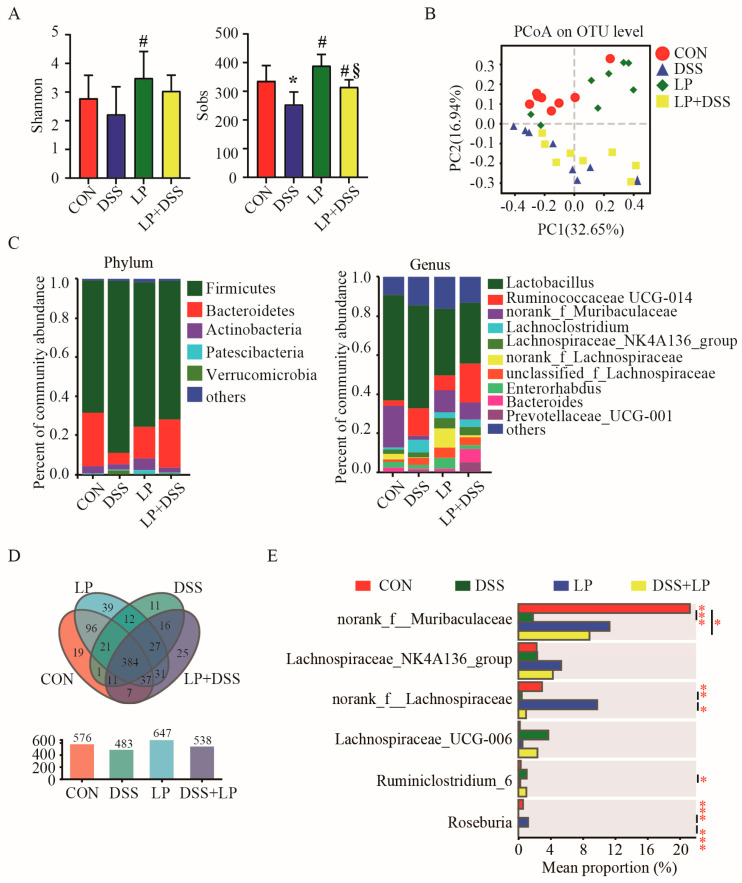
Impact of *L. Plantarum* 17-1 on gut microbiota diversity and composition in DSS-induced-colitis mice. Data are expressed as the mean ± SD (*n* = 8). Values with different superscripts are significantly different within the same level. (**A**) Shannon and Sobs index. * *p* < 0.05, compared with the control group; # *p* < 0.05, compared with the DSS group; *§ p* < 0.05, compared with the LP group. (**B**) PCoA plot based on the unweighted UniFrac distances; (**C**) microbiota composition at phylum and genus levels; (**D**) Venn diagram; (**E**) differential abundance of bacteria at genus level, * *p* < 0.05, ** *p* < 0.01, *** *p* < 0.001.

**Figure 4 nutrients-17-01348-f004:**
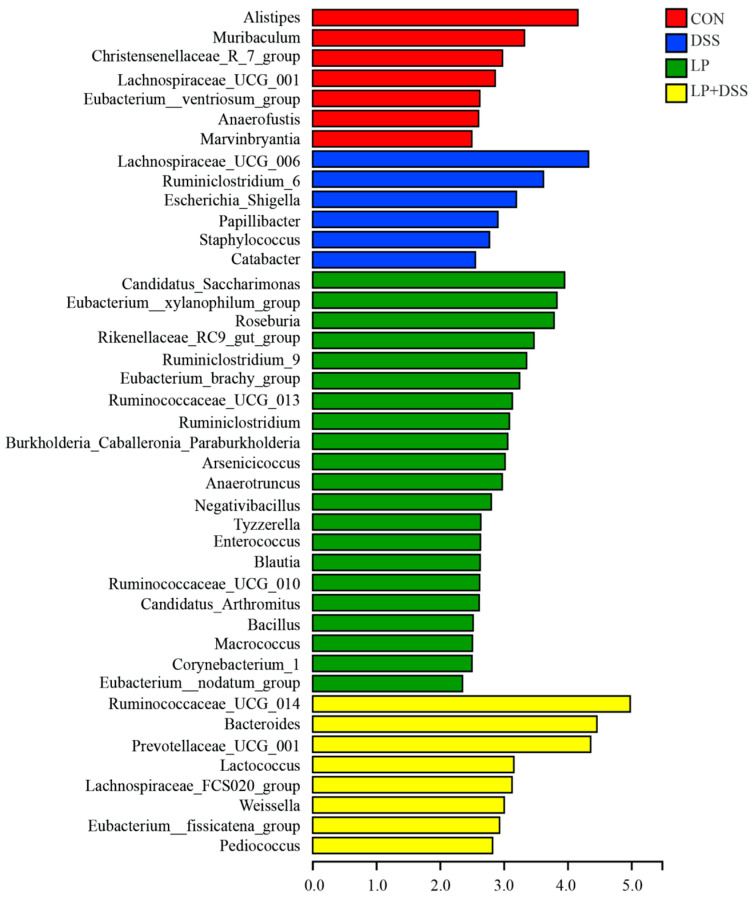
LEfSe analysis at genus level.

**Figure 5 nutrients-17-01348-f005:**
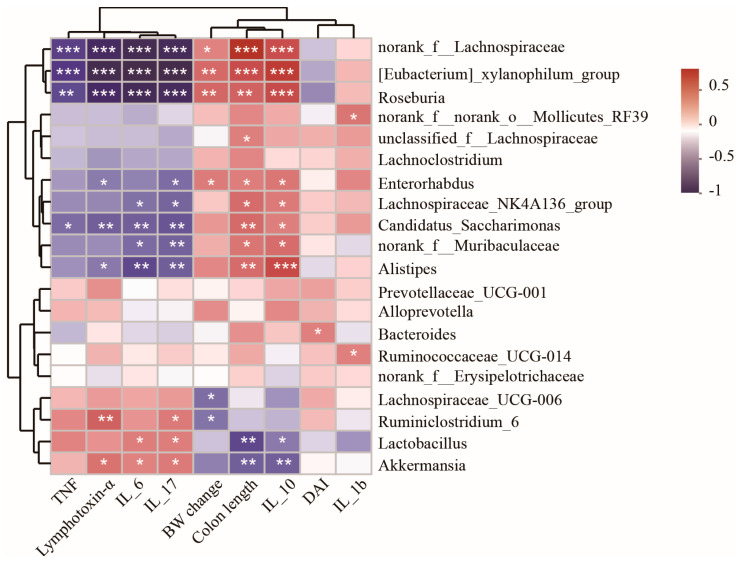
Spearman correlation analysis of differentially abundant genera and environmental factors. Values with different superscripts letters are very significantly different within the level. * *p* < 0.05, ** *p* < 0.01, *** *p* < 0.001.

**Figure 6 nutrients-17-01348-f006:**
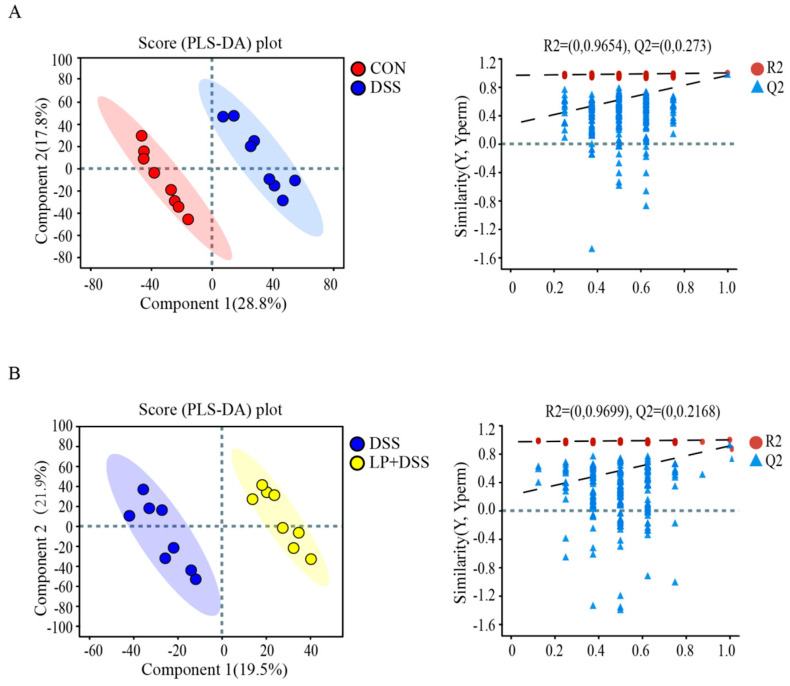
PLS-DA score plots of colonic digesta metabolites between (**A**) CON and DSS groups; (**B**) DSS and DSS + LP groups.

**Figure 7 nutrients-17-01348-f007:**
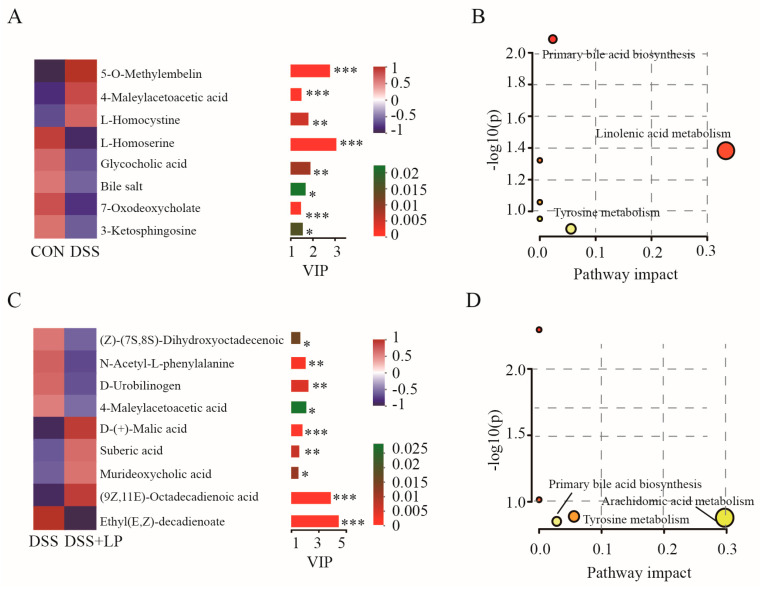
Significantly changed metabolites with VIP values above 1.5 (**A**) and significantly enriched pathways (**B**) between CON and DSS groups; significantly changed metabolites with VIP values above 1.5 (**C**) and significantly enriched pathways (**D**) between LP + DSS and DSS groups. Values with different superscripts are significantly different within the same level. * *p* < 0.05, ** *p* < 0.01, *** *p* < 0.001.

**Figure 8 nutrients-17-01348-f008:**
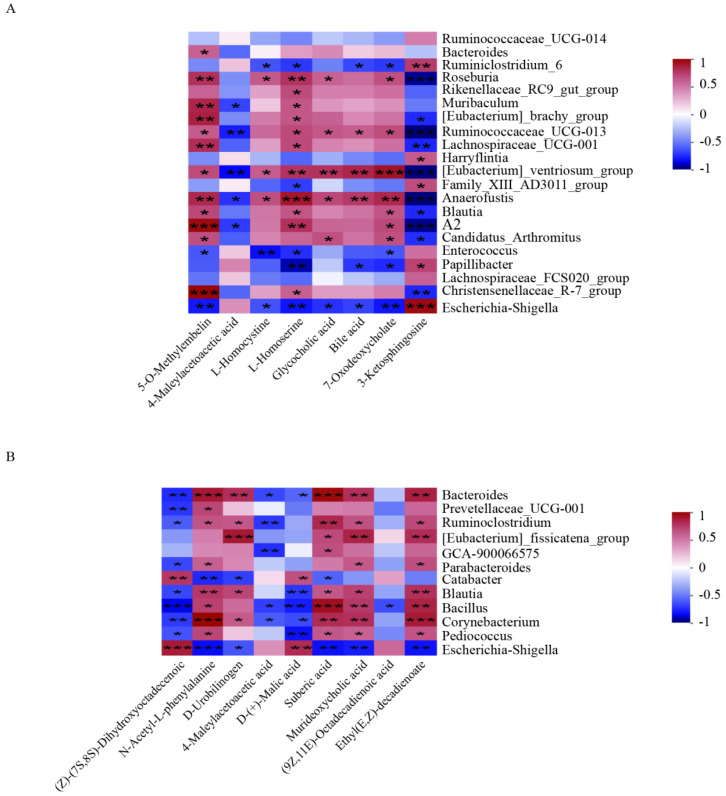
Correlation analysis of differentially abundant genera and significantly changed metabolites with VIP values above 1.5 between (**A**) CON and DSS groups; (**B**) DSS and DSS + LP groups. Values with different superscripts are significantly different within the same level. * *p* < 0.05, ** *p* < 0.01, *** *p* < 0.001.

## Data Availability

The raw reads of 16S rRNA gene sequencing were deposited into the National Center for Biotechnology Information (NCBI) Sequence Read Archive database (accession number: PRJNA1219830; http://www.ncbi.nlm.nih.gov/bioproject/1219830; NCBI, Bethesda, MD, USA). Raw metabolomics data were uploaded to the MetaboLights database (accession number: MTBLS12244; https://www.ebi.ac.uk/metabolights/reviewer48d14ea1-23ef-4c78-bfd8-8b82d3403eed; European Bioinformatics Institute, Cambridge, UK).

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
