# Peer review of "Lactobacillus plantarum 17-1 Ameliorates DSS-Induced Colitis by Modulating the Colonic Microbiota Composition and Metabolome in Mice"

_nutrients, 2025, doi:10.3390/nu17081348_

Round 1

Reviewer 1 Report

Comments and Suggestions for Authors

Review of “Lactobacillus plantarum 17-1 ameliorate DSS-induced colitis by modulating colonic microbiota composition and metabolome in mice” (nutrients-3552769)

This study investigated the utility of Lactobacillus plantarum 17-1 on DSS-induced colitis using mouse model. This study revealed that Lactobacillus plantarum 17-1 effectively suppresses inflammatory cytokines by altering inflammation-related metabolites through improvement of the gut microbiota. This study is interesting, and this reviewer has a few questions.

  1. According to the Figure 2, IL-1b level in colon tissue of LP mice was highest and that of control mice was lowest. To describe the reason for this phenomenon
  2. The ideal outcome is that the gut microbiota and metabolites that are altered in the DSS group should be the same in the LP+DSS group as in the control group. Looking only at the results, the role of gut microbiota and metabolites is completely unclear and incomprehensible to the general readers. In B and D of Fig. 7 and A and B of fig 8, the authors should explain which are linolenic acid metabolism, tyrosine metabolism, arachidonic acid metabolism and primary bile acid biosynthesis pathways, respectively.
Comments on the Quality of English Language

This reviewer is not native, thus, cannot evaluate this question.

Reviewer 2 Report

Comments and Suggestions for Authors

Ulcerative colitis is a chronic disease resulting from dysbiosis of gut microflora, intestinal epithelium damage and activation of mucosal immune system. Since this disease needs lifelong management, a dietary approach to alleviate the disease is gaining momentum. The current study deals with use of Lactobacillus plantarum as a therapeutic probiotic candidate having beneficial effects in maintaining intestinal homeostasis. The authors have made a meticulous effort towards this study.

Major comments:

Intestinal barrier damage plays a crucial role in colitis. Along with histology data, it would be useful to provide data for in vivo FD4 assay. Also, probing expression of a few tight junction associated proteins such as ZO-1, ZO-2, Claudin would definitely confirm the role of L plantarum in protecting epithelial barrier.

Comments on the Quality of English Language

There are multiple grammatical errors throughout the text. English proofreading is needed for this manuscript.

Reviewer 3 Report

Comments and Suggestions for Authors

I recommend that the authors change the Abstract (Lines 22-25). Since in this context it is difficult to understand the value of the LDA analysis, this proposal can be changed, for example: “The probiotic administration has changed the composition of the intestinal microflora and metabolome, which was confirmed by linear discriminant analysis.”

In 1998, at the 7th International TNF Congress, TNF-β was officially renamed lymphotoxin-α, and the TNF-α was renamed TNF. Please change the names of these cytokines so that they meet these recommendations.

Round 2

Reviewer 1 Report

Comments and Suggestions for Authors

The authors revised well. This reviewer has no further comments.